# Important Difference between Occupational Hazard Exposure among Shift Workers and Other Workers; Comparing Workplace before and after 1980

**DOI:** 10.3390/ijerph17207495

**Published:** 2020-10-15

**Authors:** Maud Miguet, Gull Rukh, Olga E. Titova, Helgi B. Schiöth

**Affiliations:** 1Department of Neuroscience, Functional Pharmacology, Uppsala University, 751 24 Uppsala, Sweden; gull.rukh@neuro.uu.se (G.R.); Helgi.Schioth@neuro.uu.se (H.B.S.); 2Unit of Medical Epidemiology, Department of Surgical Sciences, Uppsala University, 751 85 Uppsala, Sweden; olga.titova@surgsci.uu.se; 3Institute for Translational Medicine and Biotechnology, Sechenov First Moscow State Medical University, 119991 Moscow, Russia

**Keywords:** shift work, occupational health, work environment, hazardous substances, pollutant, world disease, epidemiology

## Abstract

Improving health and safety at work has been an important issue for the European Union since the 1980s. The existing literature supports that shift work is associated with multiple indicators of poor health but frequently neglects the potential impact of occupational hazards. This study aims at describing and comparing the exposure to different workplace hazards among shift and other workers before and after 1980. Exposure to different workplace hazards (noise, dust, pollutant, and other physical stressors) were analyzed among 119,413 participants from the UK Biobank cohort. After stratifying the analyses before and after 1980, exposure was compared between shift and other workers. Potential confounding variables (sex, age, ethnicity, education level, occupational category, and neuroticism) were adjusted for in the log-binomial regression. Shift workers had a higher prevalence ratio (PR) than other workers of being exposed to almost all identified hazards both before or after 1980. They were also more likely to be exposed to multiple hazards compared to other workers, both before 1980 (PR: 1.25; 95% CI: 1.21–1.30) and after 1980 (PR: 1.34; 95% CI: 1.30–1.38). The prevalence of all measured risk factors was higher after 1980 than before 1980 among shift workers. Of note, the work environment has improved overall for other workers. Our findings suggest that changes at the workplace have benefited other workers more than shift workers as they are still more exposed to all occupational hazards.

## 1. Introduction

High occupational exposures to physically hazardous conditions are important risk factors for the development of non-communicable diseases [1,2,3,4]. Among these hazards, there are physical factors such as noise or extreme temperatures and exposure to chemical factors such as paints, asbestos, and pesticides that are known or suspected to be related to long-term diseases [5,6,7,8]. Monitoring exposures to different physical and chemical agents should be a priority, both for improving work conditions and health.

Over the last decade, significant changes have occurred in the work environment. Due to our modern 24-h society, the number of shift workers has drastically increased [9]. In European countries, shift work is undertaken by 17 percent of the active population [10]. Several findings throughout the literature review supported that shift work is associated with multiple indicators of poor health and well-being such as a higher rate of cognitive disturbances [11], cancer [12], obesity [13], diabetes [14], and cardiovascular diseases [15]. According to the literature, this association between shift work and poorer health could be explained through the disruption of the circadian rhythms, perhaps as a result of physiological and psychological maladaptation to chronically shifted patterns [16]. Yet, since the circadian cycle is known to affect sensitivity to different stressors, exposure to hazardous substances may be worse for “non-standard hour” workers [17]. However, only few studies investigated and controlled for exposure to occupational health hazards among shift workers (e.g., noise, heat, dust, as well as chemical exposure at work). Some research has reported that shift workers are both more sensitive and more exposed to different job-related stressors [18,19,20]. For example, Parkes et al. highlighted in a cohort of 1867 oil industry personnel that shift workers have an unfavorable perception of their workplace. This was in comparison to their daytime working colleagues in the three dimensions that form the basis of the demand-control-support model of work stress [20]. Bøggild et al. found in a random sample of 5940 employees that shift workers, more often than day workers, report higher exposure to heat, annoying noise, as well as walking or standing at work [18]. In addition, consistent with reported findings, Jay et al. reported recently on a cohort of 3003 New Zealanders that shift workers were not only more likely to be exposed to all types of workplace hazards, but also more likely to be exposed to multiple hazards at one time [19].

The hazardous exposure to shift workers is frequently neglected in studies and could potentially explain the higher rates of health disorders among this population. Considering this, larger population-based epidemiological studies, including the broad variety of industries, jobs, and exposure levels, are warranted to compare the work conditions of shift workers and other workers.

Regulations on workplace standards have changed since the beginning of the century in order to offer safer and healthier work environments for employees [21,22]. According to the European Parliament, improving health and safety at work has been an important issue for the European Union since the 1980s [23]. However, most of the study designs do not investigate the trends of working conditions over time. Indeed, as mentioned by Tynes et al., changing items in repeated surveys prohibits the analysis of trends over the years [24]. Assessing this evolution is critical since work conditions are known to be a determining factor for health. The UK Biobank cohort [25] afforded the opportunity to look deeper into workplace conditions including lifetime occupational histories among more than 100,000 participants born between 1936 and 1970. A new efficient and reliable web-based tool has been developed to assess job histories [26] that enable researchers to monitor general trends of the workplace after stratifying the analysis by time.

The present paper aims to describe and compare work conditions among shift and non-shift (i.e., “other”) workers. We provide descriptive data on the numbers of UK Biobank participants working shift in different occupational categories and frequency of exposure to different risk factors. We also evaluate the trends of working conditions by comparing risk exposure before and after 1980.

## 2. Materials and Methods

The UK Biobank (UKB) is a large open source population-based prospective cohort. This study includes information from over half a million participants that were recruited between 2006 and 2010. Participants were invited to assessment centers in the United Kingdom where personal, demographic data, and basic variables were collected [27]. Ethical approval for UKB data collection was granted by the North-West Multicenter Research Ethics Committee and the use of the data at our Department was further approved by the Regional Ethics Committee of Uppsala, Sweden. All participants provided written consent with the right to withdraw at any time. Several years after joining UKB (between July and September 2015), participants were invited via email to complete the web-based occupational history questionnaire [26].


**Study population**


Overall, 122,189 participants (38%) completed the occupational history web-based questionnaire. Among them, 2760 were excluded because of incomplete data, and 16 because they withdrew consent to continue being involved in the UKB study. Based on these inclusion/exclusion criteria, 119,413 participants were included in the study.


**Primary study variables**


We were interested in the characteristics of the first paid job carried out for more than 6 months and for at least 2 days (or 15 h) a week after finishing full time education. Participants had to record their job title as well as the corresponding year to the start and end of the job. Participants were also asked about shift patterns and exposure to specific occupational hazards in this first job, as detailed below.


*Job coding*


Participants were asked to report their job by navigating down a three-level tree based on the hierarchical structure of the UK standard occupational classification version 2000 (SOC2000). According to the major job categories (i.e., 1-digit SOC codes) participants were classified into 9 categories: (1) Managers and Senior Officials, (2) Professional Occupations, (3) Associate Professional and Technical Occupations, (4) Administrative and Secretarial Occupations, (5) Skilled Trades Occupations, (6) Personal Service Occupations, (7) Sales and Customer Service Occupations, (8) Process, Plant and Machine Operatives, (9) Elementary Occupations.


*Shift pattern*


The shift pattern was assessed using a binary question (“Did you ever work shifts [day and/or night shifts] for this job?”). Day-shifts were defined as work in normal daytime hours of morning, afternoon or evening work. Night-shifts were defined as work for at least 3 h between midnight and 5am. Answers were coded as “Yes” = 1 (called “shift workers” both day and night shift included in this study) and “No” = 0 (called “other workers”).


*Year job ended*


As we were interested in the evolution of the occupational health hazards over time, we decided to split the analysis into two groups regarding the year job ended as indicated by the participant. To ensure equality of groups (numbers of participants), we split the participants from the 1980s. Hence, 59,692 participants belong to the first group (time period before 1980) and 59,721 to the second group (time period after 1980). Participants that answered “Ongoing” when data entered were part of the second group.


*Dependent variable: Occupational health hazards*


The hazardous substances exposure was evaluated from seven questions: “Thinking about the place where you worked: (1) Was it full of chemical or other fumes? (2) Was there a lot of cigarette smoke from other people smoking? (3) Was there a lot of diesel exhaust? (4) Was it very dusty? (5) Did you work with materials containing asbestos? (6) Did you work with paints, thinners or glues? and (7) Did you work with pesticides?”. Five options were available. Participants could choose one of the proposed answers (“Often”;” Sometimes”; “Rarely/Never”; “Do not know”) or they could skip answering this question. Results were transformed into a binary variable, where “never/rarely” = 0; “sometimes/often” = 1 point; and “Do not know” = missing value.

Using the same rating scale, workplace condition was assessed with three additional questions: “Thinking about the place where you worked: (8) Was it very cold? (9) Was it very hot? (10) Was it very noisy?”.


**Confounding variables**



*Demographic characteristics (sex, age, ethnic background, educational status)*


The sex variable has been acquired by the National Health Service Central Register and participants updated this information in the touch-screen questionnaire. The age variable was derived based on the date of birth and the date the participant attended an initial assessment center. Information related to ethnic background was collected during the touch-screen questionnaire in which participants answered a sequence of branching questions regarding their ethnic background. Due to the existence of multiple ethnic groups in the UKB, the original categorical variable was recoded into a dichotomous variable in which the British white ethnic group was coded as 1 and all other ethnic groups as 0. Finally, education level was recorded as “College or University degree” = 1 and other = 0. “Prefer not to answer” was coded as a missing value.


*Neuroticism*


Previous research from our team highlighted the importance of considering the confounding effect of negative personality traits for studies on the work environment [28]. Indeed, work perception may be biased by personality traits [28], and previous literature suggests shift workers to have higher neuroticism scores than other workers [29]. We therefore performed sensitivity analyses by adding neuroticism in the study model. The neuroticism score was based on 12 neurotic behavior domains from the touchscreen questionnaire at baseline. The range scored between 0 and 12 with a higher value indicating higher neuroticism level.


**Statistical analyses**


Statistical analyses were carried out with SPSS software (IBM SPSS Statistics version 26). The frequency (number and percentage) of those who worked shift in each job category (using SOC 2000 coding) was calculated after stratifying for job period, before and after 1980. The odds ratio by logistical regression can overestimate the prevalence ratio in cross-sectional studies, especially when working with frequent outcomes (above 10%) [30]. Therefore, we adopted a prevalence ratio (PR) calculated by the log-binomial regression method to estimate the risk of being exposed to occupational health hazards (noise, cold, hot, dust, chemical or other fumes, smoke from other people smoking, asbestos, paints/thinners or glues, pesticides, and diesel exhaust) among shift workers compared to other workers (reference). The PR represent the risk of being exposed to the different occupational health hazards, for example, a PR of 2 for diesel exhaust can be interpreted to mean that the proportion of workers being exposed to diesel exhaust is 2-fold higher among shift compared to non-shift workers. In a similar manner, log-binomial regression analysis was used to compare the evolution of workplace perception before 1980 (reference) and after 1980, both separately for shift and other workers. All the analyses were adjusted for sex, ethnicity, education level, age, and occupational category. The potential impact of personality traits has been considered in the sensitivity analyses, by additionally adjusting the analyses for the neuroticism score. The PR and 95% confidence intervals (CIs) were determined by the log-binomial regression model [31]. We considered correction for multiple testing (Bonferroni correction) and set the critical p level for significance at 0.05/10 = 0.005 since we tested 10 hazards. Missing data were excluded for the analyses and the percentages of missing values are reported in Appendix A. Graphs were prepared using GraphPad Prism version 8.1.2.

## 3. Results

We undertook analyses among 119,413 participants distributed in two groups according to the year job ended, either before or after 1980. The percentage of shift workers was 11.4% (*n* = 6800) in the first period and 17.5% (*n* = 10,439) in the second one. Mean age of the participants at the recruitment were respectively 59.6 (±5.4) and 52.6 (±8.0) years. Study descriptions and distribution in the different job categories are provided in Table 1. As illustrated in Figure 1, the percentage of shift workers increased in each job category after 1980, with a higher increase being found among “Process, Plant, and Machine Operatives” (+21%), followed by “Skilled Trades Occupations” (+14.2%) and the “Elementary Occupations” (+11.2%). The distribution of shift work across the different job categories highlighted “Administrative and Secretarial Occupations”, “Sales and Customer Service Occupation” and “Professional Occupation” as the sectors with the lowest numbers of people working shift.

### 3.1. Hazard Exposure

Prevalence of workplace hazards is reported in Table 2 and Table 3. Table 2 allows comparison between shift and other workers (either before or after 1980), whereas Table 3 highlights changes over time in each group (i.e., changes among shift workers, and changes among other workers in the period before and after 1980).

#### 3.1.1. Comparison between Shift and Other Workers

Before 1980, participants working shift were more likely to be exposed to all hazardous conditions than other workers, except pesticides (PR [95% CI]: 0,98 [0.81–1.17]) and smoke from other people smoking (PR [95% CI]: 0.89 [0.87–0.91]) (Table 2 and Figure 2A).

After 1980, participants working shift had statistically significant higher prevalence than other workers of being exposed to each of the reported workplace hazards (noise: 1.57 [1.52–1.61]; cold: 1.19 [1.26–1.32]; hot: 1.39 [1.37–1.42]; dust: 1.17 [1.14–1.21]; chemical or other fumes: 1.81 [1.75–1.88]; smoke from other people smoking: 1.04 [1.02–1.06]; asbestos: 1.54 [1.44–1.64]; paints/thinners or glues: 1.19 [1.13–1.26]; pesticides: 1.15 [1.01–1.32]; and diesel exhaust: 2.12 [2.01–2.23]) (Table 2 and Figure 2B).

#### 3.1.2. Comparison Between Before 1980 and After 1980

Among the shift workers, the prevalence of being exposed to noise, cold, hot, dust, chemical or other fumes, pesticides, and diesel exhaust were significantly higher after 1980 comparing before 1980 (Table 3 and Figure 3A). Exposure to paint/thinner/glue and asbestos were unchanged, whereas exposure to passive smoking was significantly reduced. After Bonferroni adjustment, dust exposure was not significantly different anymore (*p* = 0.019).

Among the other workers, the prevalence for noise, cold, and hot were increased, whereas exposure to dust: 0.94 [0.92–0.97]; chemical or other fumes: 0.78 [0.75–0.81]; smoke from other people smoking: 0.79 [0.78–0.80]; asbestos 0.94 [0.89–0.99]; and diesel exhaust: 0.88 [0.83–0.93] were decreased. There were no statistically significant differences among prevalence for paints/thinners or glues and pesticides exposure for other workers after 1980 (Table 3 and Figure 3B). After Bonferroni adjustment, asbestos exposure was not significantly different anymore (*p* = 0.015).

#### 3.1.3. Multiple Hazards Exposure

Shift workers were more likely to be exposed to multiple hazards (two or more) in the workplace compared to those working non-shift (Figure 4), either before 1980 (PR: 1.27; 95% CI: 1.25–1.30) or after 1980 (PR: 1.35; 95% CI: 1.33–1.38) (Table 2).

### 3.2. Sensitivity Analyses

Further adjustment for the neuroticism trait is reported in the sensitivity analyses. Appendix A allows comparison between shift and other workers stratifying for job period. In line with the main analyses, we found that before 1980, participants working shift were more likely to be exposed to all hazardous conditions than other workers, except pesticides and smoke from other people smoking. After 1980, participants working shift had statistically significant higher prevalence than other workers of being exposed to each of the reported workplace hazards. Appendix A highlights changes over time in each group. Similar to the main analyses, among the shift workers, all the risk factors were significantly more prevalent after 1980 comparing before 1980, except smoke and paint/thinner or glue exposure. Among the other workers, the prevalence for noise, cold, hot, and paints/thinners or glues exposure were increased, whereas exposure to dust, chemical or other fumes, smoke from other people smoking, asbestos, and diesel exhaust were decreased.

## 4. Discussion

The present study provided the opportunity to compare the work environment of shift and other workers in two different time periods. Our findings revealed that shift workers report higher exposure to hazardous conditions than other workers, both before 1980 and after 1980. Moreover, while the work conditions seem to have been overall improved for all workers, our results highlight the potential of higher exposure to hazards after 1980, compared with before 1980, for the shift workers.

This higher exposure to different occupational risk factors among shift workers compared with other workers could perhaps in part explain the higher prevalence of non-communicable diseases reported in this population [29]. It is possible that previous studies, often focusing on the behavior of shift workers, may have underestimated exposure-related factors that could contribute to increased risks for diseases. It has been previously demonstrated that several physical aspects of the workplace environment (e.g., noise, temperature and pollutant exposure) may have an impact on the development of serious adverse metabolic and mental health diseases [4]. The discrepancies between shift and other workers in occupational exposure could thereby contribute to the established association between poor health and shift work.

In accordance with previous literature [13], our results supported that shift workers were not only more likely to be exposed to several types of workplace hazards, but also more likely to be exposed to multiple hazards at once. This issue is of high relevance since in some cases, the overall effect of multiple hazardous exposure is considerably greater than the individual effects, as there may be synergistic relationships between the different hazards [32]. Again, this result could possibly explain, at least in part, the unfavorable health conditions reported among shift workers [29].

The differences in exposure to hazardous conditions reported by shift workers and other workers could be explained by the nature of the jobs involved. Indeed, according to both our results and previous literature, shift work is more common in some occupations (e.g., industrial sectors, manufacturing, and transport) and these industries have often weaker occupational health and safety conditions, and increased exposure to hazardous substances, noise, and manual handling risks [33]. In addition, evening and night work are highly prevalent among protective services (police, security guards) and these employees, particularly those that are working outdoors, are exposed to different types of weather, including uncomfortable temperatures. This point suggests that shift work is more prevalent among jobs with unfavorable working environments. Yet, even when controlling for social class and job coding, our results still show higher prevalence of hazardous conditions among shift workers than other workers.

Some authors suggest that the higher exposure risk reported by shift workers is due to an altered perception induced by specific personality traits [34]. Indeed, evidence suggests that shift workers differ from day workers in negative affectivity [29]. Moreover, previous research by our team highlighted the neuroticism trait to be an important confounding effect in work perception [28]. Our results are based on declarative information and thereby reflect the interaction between the real working environment and its perception by the worker. However, we performed further adjustments to the neuroticism score and did not find substantially different results. More specifically, we have not found here that further adjustment on personality traits affect the association between occupational exposure and shift pattern.

Our analysis shows that the prevalence of shift work has drastically increased, from 11.4% to 17.5% after 1980. While previous studies also reported an important increase in the prevalence of shift work [10], we did not find other papers addressing the evolution of occupational exposure over time among this population. Efforts to improve worker safety were developed since the beginning of the century (such as improved ventilation and dust suppression, for example) leading to the overall reduction of occupational health risks and workplace deaths [23,35]. However, most of the previous study designs have not enabled studies of this kind and it has therefore been difficult to draw conclusions of how shift work has changed over time. Contradictory with what we might have expected, our results indicate that the prevalence of being exposed to different risk factors was higher after 1980 than before 1980 for the shift workers, while the work environment has been overall improved for other workers (lower prevalence of being exposed to dust, chemical or other fumes, smoke from other people smoking, asbestos, and diesel exhaust after 1980 than before 1980). Our findings could suggest that changes at the workplace may have benefited some workers more than others. In this regard, it could perhaps be advisable that authorities ensure protective threshold limit values, not only for “normal working conditions”, but also specifically for “non-standard hour workers”, like shift work personnel.

The strength of this analysis is that it includes a large panel of participants, involving a variety of education levels, job categories, socioeconomic status, in contrast to most previous studies that include a single-workplace or small cohorts. We performed sensitivity analyses adjusting for neuroticism, which is relevant to the interpretation of the findings. This suggests that the variance in occupational hazards exposure is not due to a personality trait or a self-report bias. Moreover, our study design also brought very relevant information about the general trends in the work environment which are rarely investigated in other studies. Yet, it is important to note the limitations of this study, which highlights perspectives for further work. Firstly, information regarding exposure to risk factors was retrospectively and subjectively recorded and was only sought on the presence or absence of the hazards, not on quantitative exposure levels or timing of exposure which limits us to judge the magnitude of the risk of disease development. In addition, we may have considered other potential confounders in the analyses such as sleep parameters since shift workers are known to sleep less than day workers and to be more tired [36], which could have affected the perception of the work environment. However, we did not have access to retrospective information regarding sleep quality. Lastly, the present study did not directly examine health outcome, so the link between exposure to occupational health hazards, shift work, and poorer health remains to be explored. Further work is required to deeply investigate the relation between health, occupational exposure to hazards and shift pattern. Specifically, studies are needed to analyze more specific work patterns (day or night shit, duration of the shifts, rest day during the shift period, etc.).

## 5. Conclusions

In conclusion, this study found that whereas significant improvements have occurred among non-shift workers’ environment, workplace safety among shift workers remains insufficient. We found higher risks of being exposed to all occupational hazardous conditions among shift workers compared with non-shift workers. Addressing the possibly damaging effects of the work environment is a critical element when considering the multitude of potential factors that may contribute to the adverse health status reported among shift workers. The present data suggest that occupational health exposure among shift workers may contribute to their adverse health status, compared with other workers. In this respect, the present study adds to the existing shift work literature and aims to call attention to occupational health exposure among this specific population. Future studies have to consider that exposure to hazardous conditions may differ among shift workers and non-shift workers.

## Figures and Tables

**Figure 1 ijerph-17-07495-f001:**
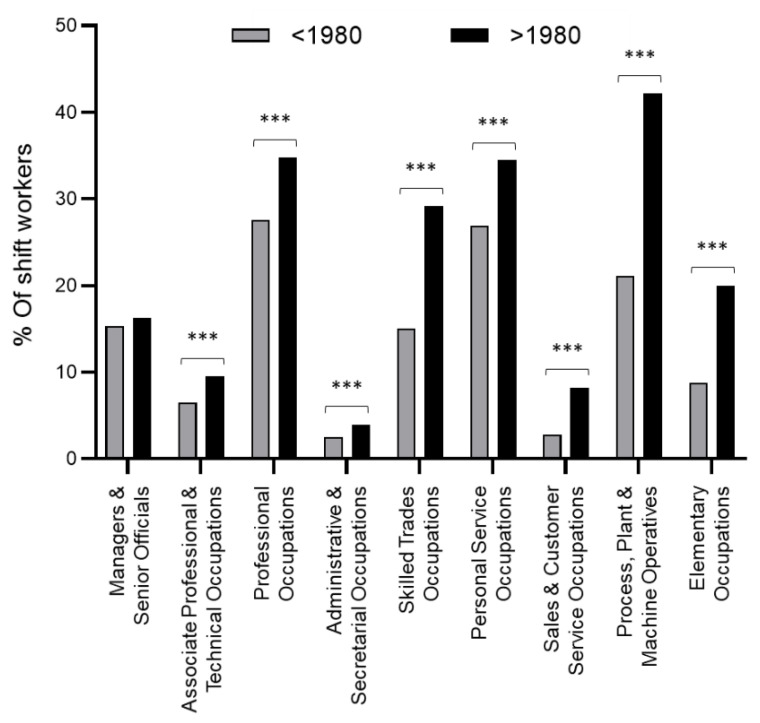
Percent of shift workers among job categories before and after 1980. ***: *p*-value < 0.001.

**Figure 2 ijerph-17-07495-f002:**
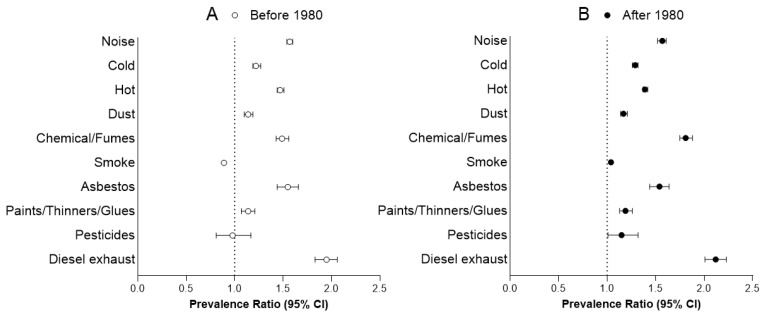
Prevalence ratio of occupational risk exposure among shift workers compared with other workers. (**A**) Prevalence for shift workers compared to other workers (reference) before 1980; (**B**) Prevalence for shift workers compared to other workers (reference) after 1980. Analysis controlled for sex, ethnicity, education level, age, and occupational category.

**Figure 3 ijerph-17-07495-f003:**
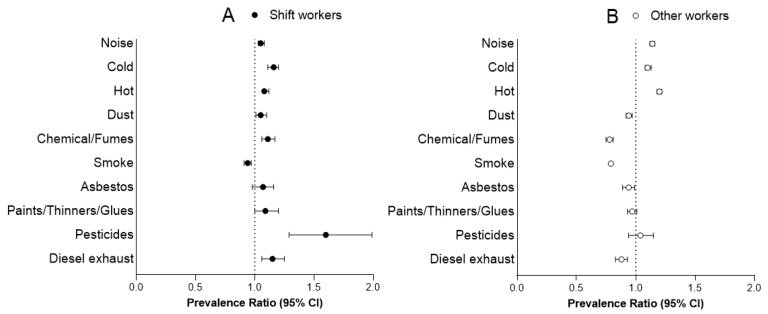
Prevalence ratio of occupational risk exposure, exposure after 1980 compared with before 1980. (**A**) Prevalence for shift workers after 1980, compared to shift workers before 1980 (reference); (**B**) Prevalence for other workers after 1980, compared to other workers before 1980 (reference). Analysis controlled for sex, ethnicity, education level, and age.

**Figure 4 ijerph-17-07495-f004:**
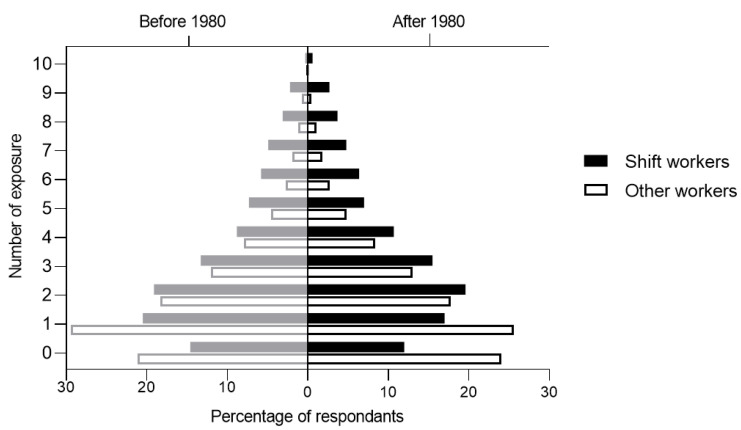
Number of workplace hazards comparing shift and other workers.

**Table 1 ijerph-17-07495-t001:** Sample characteristics.

	Shift Workers *n* = 17,239	Non-Shift Worker *n* = 102,174
Sex—Female	46.3 (7986)	57.2 (58,468)
Ethnicity—White British	89.8 (15,428)	92.2 (93,857)
Education—College or University	42.4 (7248)	48.3 (48,991)
Age (Mean ± Standard Deviation)	55.2 ± 7.8	56.2 ± 7.6
Managers and Senior Officials	1361 (7.9)	7125 (7.0)
Professional Occupations	2898 (16.8)	32,004 (31.3)
Associate Professional and Technical Occupations	7018 (40.7)	15,302 (15.0)
Administrative and Secretarial Occupations	772 (4.5)	24,955 (24.4)
Skilled Trades Occupations	1817 (10.5)	6.650 (6.5)
Personal Service Occupations	1143 (6.6)	2.632 (2.6)
Sales and Customer Service Occupations	237 (1.4)	4768 (4.7)
Process, Plant and Machine Operatives	1089 (6.3)	2692 (2.6)
Elementary Occupations	904 (5.2)	6046 (5.9)

**Table 2 ijerph-17-07495-t002:** Prevalence ratio of workplace hazards among shift workers compared to other workers (reference).

	Job Ended Before 1980 *n* = 59,692						Job Ended After 1980 *n* = 59,721					
Exposure	% Shift Workers	% Other Workers	PR	95% CI		*p* Value	% Shift Workers	% Other Workers	PR	95% CI		*p* Value
*Noise*	64.0	39.6	1.57	1.54	1.60	<0.001	70.3	43.4	1.57	1.52	1.61	<0.001
*Cold*	35.3	27.0	1.22	1.19	1.27	<0.001	43.7	29.7	1.29	1.26	1.32	<0.001
*Hot*	56.7	36.2	1.47	1.44	1.51	<0.001	64.3	43.4	1.39	1.37	1.42	<0.001
*Dust*	29.8	25.2	1.14	1.10	1.19	<0.001	32.2	23.9	1.17	1.14	1.21	<0.001
*Chemical/fumes*	26.2	16.1	1.49	1.43	1.56	<0.001	29.5	13.8	1.81	1.75	1.88	<0.001
*Smoke*	48.4	53.2	0.89	0.87	0.91	<0.001	45.5	40.3	1.04	1.02	1.06	0.001
*Asbestos*	13.6	7.1	1.55	1.44	1.66	<0.001	13.1	5.9	1.54	1.44	1.64	<0.001
*Paints/thinners/glues*	13.2	10.2	1.14	1.07	1.21	<0.001	15.2	10.5	1.19	1.13	1.26	<0.001
*Pesticides*	2.0	1.8	0.98	0.81	1.17	0.79	2.8	2.0	1.15	1.01	1.32	0.037
*Diesel exhaust*	16.5	6.8	1.95	1.83	2.06	<0.001	20.1	6.5	2.12	2.01	2.23	<0.001
*Exposed to ≥2hazards*	64.8	49.5	1.25	1.21	1.30	<0.001	71.0	50.2	1.34	1.30	1.38	<0.001

Analysis were adjusted for gender, ethnicity, education, age, occupational category and stratified by time. PR: Prevalence ratio; CI: confidence interval; *p* value: probability value.

**Table 3 ijerph-17-07495-t003:** Prevalence ratio of workplace hazards after 1980 compared to before 1980 (reference).

	Shift Workers *n* = 17,239						Other Workers *n* = 102,174					
Exposure	% before 1980	% after 1980	PR	95% CI		*p* Value	% before 1980	% after 1980	PR	95% CI		*p* Value
*Noise*	64.0	70.3	1.05	1.03	1.08	<0.001	39.6	43.4	1.14	1.12	1.16	<0.001
*Cold*	35.3	43.7	1.16	1.11	1.20	<0.001	27.0	29.7	1.10	1.08	1.13	<0.001
*Hot*	56.7	64.3	1.08	1.06	1.12	<0.001	36.2	43.4	1.20	1.18	1.22	<0.001
*Dust*	29.8	32.2	1.05	1.01	1.10	0.019	25.2	23.9	0.94	0.92	0.97	<0.001
*Chemical/fumes*	26.2	29.5	1.11	1.06	1.17	<0.001	16.1	13.8	0.78	0.75	0.81	<0.001
*Smoke*	48.4	45.5	0.94	0.91	0.97	<0.001	53.2	40.3	0.79	0.78	0.80	<0.001
*Asbestos*	13.6	13.1	1.07	0.98	1.16	0.118	7.1	5.9	0.94	0.89	0.99	0.015
*Paints/thinners/glues*	13.2	15.2	1.09	1.00	1.20	0.054	10.2	10.5	0.97	0.93	1.01	0.161
*Pesticides*	2.0	2.8	1.60	1.29	1.99	<0.001	1.8	2.0	1.04	0.94	1.15	0.475
*Diesel exhaust*	16.5	20.1	1.15	1.06	1.25	0.001	6.8	6.5	0.88	0.83	0.93	<0.001
*Exposed to ≥2hazards*	64.8	71.0	1.04	1.01	1.06	0.003	49.5	50.2	1.04	1.02	1.05	<0.001

Analysis were adjusted for gender, ethnicity, education, age and stratified by shift pattern. PR: Prevalence ratio; CI: confidence interval; *p* value: probability value.

## Data Availability

All data supporting this study is openly available from the UK Biobank (https://www.ukbiobank.ac.uk/).

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
