# Peer review of "Important Difference between Occupational Hazard Exposure among Shift Workers and Other Workers; Comparing Workplace before and after 1980"

_ijerph, 2020, doi:10.3390/ijerph17207495_

Round 1

Reviewer 1 Report

The paper has been written about a very important issue of health and safety especially among shift workers. The aim of the paper is very clear, and the findings from the study have been clearly presented. Equally the style of writing is consistent and more importantly the references used for the study are appropriate, relevant and current. However, two critical omission from the study make the originality, soundness and appeal to readers less than optimal. one of the omissions is the strong motivation of the cut-off year of 1980. The authors do not offer any convincing explanation why the year 1980 was picked for the study. Much more should be included in the study as opposed to just saying " Improving health and safety at work has been an important issue for the European Union 17 since the 1980s". The second omission is the reference to other studies that have looked at the exposure levels of the two different worker groups. This aspect is not clear on what is already known about the hazard exposure levels of workers.

Author Response

The paper has been written about a very important issue of health and safety especially among shift workers. The aim of the paper is very clear, and the findings from the study have been clearly presented. Equally the style of writing is consistent and more importantly the references used for the study are appropriate, relevant and current. However, two critical omission from the study make the originality, soundness and appeal to readers less than optimal. one of the omissions is the strong motivation of the cut-off year of 1980. The authors do not offer any convincing explanation why the year 1980 was picked for the study. Much more should be included in the study as opposed to just saying " Improving health and safety at work has been an important issue for the European Union 17 since the 1980s". The second omission is the reference to other studies that have looked at the exposure levels of the two different worker groups. This aspect is not clear on what is already known about the hazard exposure levels of workers. 

We thank the reviewer for these pertinent comments. We have corrected a typo in the methods which we think was the source of the confusion. In accordance with the reviewer, it has been precised in the revised version of the paper that the cut-off year of 1980 was motivated to obtain equal group number.

Secondly, although we were referring to some important studies that framed our work in the introduction, we agree with the reviewer that some relevant details regarding the previous literature were missing, this has been edited in the revised version of the paper, as requested.

Reviewer 2 Report

The problem analysis is clearly described. The set-up of the work makes a professional impression. The structure of the paper is fine.    For the general reader it is recommended to define the prevalence ratio and illustrate it with an example.   Page 2 Line 53: Can you give examples of other causes why shift workers have a higher rate of disorders?   Page 3 Line 116-117: the two samples of before and after 1980 are about the same size. Is that coincidentally or by purpose?   Page 4 Line 165: What do you exactly mean with "adjusting" for sex and other?    Line 167: And also with controlling the analyses for the neuroticism score? This is not immediately clear.   Page 11 line 345/346: What exactly do you mean with: "Future studies have to consider that exposure to hazardous conditions may differ among shift workers and non-shift workers."? Is it that future studies must take your results into account, or do you mean something different?   There are a few linguistic flaws in verb tense, e.g., on page 3: assess, transform and attending, while group is groups.

Author Response

The problem analysis is clearly described. The set-up of the work makes a professional impression. The structure of the paper is fine. For the general reader it is recommended to define the prevalence ratio and illustrate it with an example.

We thank the reviewer for this comment. As suggested, we added a definition and an example to illustrate the prevalence ratio, which makes it easier to understand.

Page 2 Line 53: Can you give examples of other causes why shift workers have a higher rate of disorders?  

As requested by the reviewer, we developed this aspect in the introduction.

Page 3 Line 116-117: the two samples of before and after 1980 are about the same size. Is that coincidentally or by purpose?  

This has been detailed in response to the first reviewer: the cut-off year of 1980 was motivated to obtain equal group number. This has been clarified in the revised version of the manuscript.

Page 4 Line 165: What do you exactly mean with "adjusting" for sex and other?    Line 167: And also with controlling the analyses for the neuroticism score? This is not immediately clear.  

We agree with the reviewer that it was not clear. Accordingly, we reformulated it consistently in the revised manuscript.

Page 11 line 345/346: What exactly do you mean with: "Future studies have to consider that exposure to hazardous conditions may differ among shift workers and non-shift workers."? Is it that future studies must take your results into account, or do you mean something different?  

Indeed, exposure to unfavorable work characteristic plays a significant role in the development of non-communicable disease and this could potentially explain the higher rate of disorders among the shift workers.

There are a few linguistic flaws in verb tense, e.g., on page 3: assess, transform and attending, while group is groups.

We thank the reviewer for his careful reading. This has been corrected.

Reviewer 3 Report

This manuscript aims to show the differences in occupational hazard exposure between shiftworkers and non-shiftworkers using the UK biobank data.  It is complicated by the fact that they also try and look at the differences before and after 1980.  It would be a cleaner and more useful paper if they used the most current job, and just compared workers by their shift status (see comment for line 109).  They could then do a more comprehensive job of looking at exposure to hazards over time in a separate paper using more time categories rather than two very broad time groups.

Considerable English editing is required.

Line 40: should be physical and chemical agents

Line 45: The authors imply that “the literature review” comprehensively supports that shiftwork is associated with 5 major medical conditions.  The references 11 to 16 do not support this statement, being 2 studies from the biobank cohort (metabolic, mental health and diabetes), 1 from another cohort (one psychological test), two reviews (CVD and vascular events) and a very old review on health disorders of shift workers.  It does not look like the authors have a broad and detailed knowledge of the shiftwork and health literature.

Line 51:  none of the 3 papers cited have anything about shiftworkers being more sensitive to workplace stressors.

Line 61: this paper does not state that “most of the studies design do not allow to study the trends of working condition over time”.  It just mentions that often the items change over time making it more difficult to look at trends.

Line 65: this paper does not mention assessing job histories, it is an innovative way to code job histories to SOC codes.

Line 74: The authors should mention that the data were obtained from the open source - currently it could be understood as they have collected it themselves.

Line 109: To include day shift workers in the category of “shift workers” is very strange.  The authors should analyse the night workers separately, in 4 categories similar to what was done in Wyse et al 2017.

Line 113: what perception are the authors talking about?

Line 115 What “special value” was used? How were these participants categorized?  Using only 2 categories with a random cut point limits the usefulness of these 

Line 119: Were these questions asked for each job?

Line 137: Given that the job chosen was more than 40 years previously for half of the participants, why was age at the time of recruitment used for the analysis?  The age for the earlier time period is obviously going to be older (line 178), but does not relate to the age they were when doing the job.

Line 146: The references provided suggest that those who score higher on neuroticism scores also report worse psychological conditions at work.  But is there any evidence that they report worse physical conditions?

Table 1.  Given that the interest is in the association between shiftwork and working conditions, the columns in this table should be shiftwork Y/N (or better still, the 4 categories of shiftwork as in Wyse et al). 

Figure 1.  Since the percent of shiftworkers differs by occupation (Figure 1) and we know that the percent of workers exposed to hazards differs by occupation, the PRs in Table 2 and Figure 3 should be adjusted for occupation.  Otherwise all the differences between shift workers and non-shiftworkers might just be due to confounding by occupation.

Author Response

This manuscript aims to show the differences in occupational hazard exposure between shiftworkers and non-shiftworkers using the UK biobank data.  It is complicated by the fact that they also try and look at the differences before and after 1980.  It would be a cleaner and more useful paper if they used the most current job, and just compared workers by their shift status (see comment for line 109).  They could then do a more comprehensive job of looking at exposure to hazards over time in a separate paper using more time categories rather than two very broad time groups.

We thank the reviewer for his assessment. Accordingly, we revised the manuscript and have added precision throughout the introduction, the methods and the discussion, which clearly improved our paper and its understanding.

Considerable English editing is required.

The manuscript has been proofread with careful attention towards English editing.

Line 40: should be physical and chemical agents

We thank the reviewer for this comment; the suggested correction has been made.

Line 45: The authors imply that “the literature review” comprehensively supports that shiftwork is associated with 5 major medical conditions.  The references 11 to 16 do not support this statement, being 2 studies from the biobank cohort (metabolic, mental health and diabetes), 1 from another cohort (one psychological test), two reviews (CVD and vascular events) and a very old review on health disorders of shift workers.  It does not look like the authors have a broad and detailed knowledge of the shiftwork and health literature.

As requested by the reviewer, we updated the references to be more relevant and current.

Line 51:  none of the 3 papers cited have anything about shiftworkers being more sensitive to workplace stressors.

Once more, we thank the reviewer for this comment which led us to develop this paragraph in the revised version of the manuscript. Importantly, Parkes et al. highlighted that “shiftworkers perceived their environment significantly less favorably than dayworkers”; in a consistent manner, Bogglid et al. reported that “Shift workers more often than day workers report higher exposure to work environment” and finally Jay et al. found that “Participants working non-standard hours are more likely to report exposure to all identified hazards in their workplace”.

Line 61: this paper does not state that “most of the studies design do not allow to study the trends of working condition over time”.  It just mentions that often the items change over time making it more difficult to look at trends.

In accordance with the reviewer, the sentence has been reformulated.

Line 65: this paper does not mention assessing job histories, it is an innovative way to code job histories to SOC codes.

We agree with the reviewer that the sentence was not well-formulated. The sentence has been revised.

Line 74: The authors should mention that the data were obtained from the open source - currently it could be understood as they have collected it themselves.

As requested by the reviewer, this has been mentioned in the revised version of the paper.

Line 109: To include day shift workers in the category of “shift workers” is very strange.  The authors should analyse the night workers separately, in 4 categories similar to what was done in Wyse et al 2017.

We agree with the reviewer that this is an interesting point that should be discussed, and expressed it in the discussion. There is a need for further studies, specifically questioning the potential characteristic of specific shift pattern (day or night shit, duration of the shifts, rest day during the shift period, etc.). The design of the present study was based on the number of participants available in each group. Indeed, if we consider the night workers separately, we end-up with n = 159 night workers before 1980, and n = 404 night workers after 1980 which decrease the interest of such grouping.

Line 113: what perception are the authors talking about?

As requested by the reviewer we have clarified this sentence.

Line 115 What “special value” was used? How were these participants categorized?  Using only 2 categories with a random cut point limits the usefulness of these 

We agree with the reviewer; these participants belong to the second group. We reformulate this in the manuscript.

Line 119: Were these questions asked for each job?

We thank the reviewer for this question. This has been detailed in the methods

Line 137: Given that the job chosen was more than 40 years previously for half of the participants, why was age at the time of recruitment used for the analysis?  The age for the earlier time period is obviously going to be older (line 178), but does not relate to the age they were when doing the job.

We agree with the reviewer that this could be a delicate point. As our variable of interest (i.e. exposure to occupational hazard) is a retrospective measurement, we considered the age of the participants at the time of recruitment in the study, and not the age at the start of job.

Line 146: The references provided suggest that those who score higher on neuroticism scores also report worse psychological conditions at work.  But is there any evidence that they report worse physical conditions?

Although this question merits further consideration, there is strong evidence that individuals with high neuroticism score perceive their environment in a negative light. Therefore, we can formulate, in light of the existing literature, the hypothesis that they also report worse physical conditions.

Table 1.  Given that the interest is in the association between shiftwork and working conditions, the columns in this table should be shiftwork Y/N (or better still, the 4 categories of shiftwork as in Wyse et al). 

We thank the reviewer for his precious advice. The table has been updated.

Figure 1.  Since the percent of shiftworkers differs by occupation (Figure 1) and we know that the percent of workers exposed to hazards differs by occupation, the PRs in Table 2 and Figure 3 should be adjusted for occupation.  Otherwise all the differences between shift workers and non-shiftworkers might just be due to confounding by occupation.

We totally agree about this statement regarding further adjustment for occupational category. According to this pertinent comment, we redid the analysis again and changed the results (figure, table + text) accordingly.

Reviewer 4 Report

This study aims at investgating on the exposure to different workplace hazards among shift and other workers before and after 1980. Exposure to different workplace hazards (noise, dust, pollutant and so on) were studied in 119,413 participants. As potential confounding variables, sex, age, ethnicity, education level, occupational category and neuroticism were controlled. The findings suggest that shift workers are still more exposed to the considered occupational hazards.

On the whole, I have a positive opinion on this study.

Only few minor points:

  1. page 2, rows 45-47: please add, sleepiness and sleep disturbances within the list of consequences of shift work (Di Muzio et al., Front. Neurosci. | doi: 10.3389/fnins.2020.579938)

  1. The are some limitations that should be cited in the discussion:
  • Since the shift pattern was assessed using a binary question (“Did you ever work shifts (day and/or 117 night shifts) for this job?”), this implies that the study misses the possibility to evaluate the effects of different shift work schedules
  • As potential confounders in the analyses, no sleep parameters or measures have been considered

Author Response

We thank the referee for the careful review of our manuscript.

Accordingly, we added the relevant reference in our manuscript.

Moreover, we agree with the reviewer and discussed the two limitations  (lines 347-349 and lines 352-355).